# A Comparison of the Effects of Short-Term Physical and Combined Multi-Modal Training on Cognitive Functions

**DOI:** 10.3390/ijerph19127506

**Published:** 2022-06-19

**Authors:** Claudia Kardys, Kristina Küper, Stephan Getzmann, Michael Falkenstein, Claudia Voelcker-Rehage

**Affiliations:** 1Institute for Health & Social Affairs, FOM University of Applied Sciences, D-45127 Essen, Germany; claudia.kardys@fom.de; 2Bundeswehr Institute for Preventive Medicine, D-56626 Andernach, Germany; kristinakueper@bundeswehr.org; 3Leibniz Research Centre for Working Environment and Human Factors, Technical University of Dortmund, D-44139 Dortmund, Germany; 4Institute for Working, Learning and Aging (ALA), D-44805 Bochum, Germany; falkenstein@ala-institut.de; 5Department of Neuromotor Behavior and Exercise, Institute of Sport and Exercise Sciences, University of Muenster, D-48149 Muenster, Germany; claudia.voelcker-rehage@uni-muenster.de

**Keywords:** short-term intervention, multi-modal training, physical training, motor–cognitive performance, cognitive functions

## Abstract

Physical training has beneficial effects not only on physical fitness, but also on cognitive functions. The most effective way to improve cognitive functions via physical training as well as the degree to which training effects transfer to untrained cognitive functions is still unclear, however. Here, we investigated the effects of adaptive and multi-modal short-term training interventions on cognitive training gains and transfer effects. Over a period of 12 weeks, 102 employees of a car manufacturing company (age range 20 to 61 years) received trainer-guided exercises, consisting of either two adaptive training interventions, physical (strength) training and multi-modal (motor–cognitive) training, or non-adaptive strength training (active control group). For the multi-modal intervention, the “Agility Board” was employed, a novel, multi-modal training device. Pre- and post-training, psychometric tests were conducted to measure cognitive abilities, such as perceptual speed, attention, short-term memory, working memory, inhibition, and mental rotation. In addition, motor–cognitive performance was assessed. Compared with the active control group, both training groups showed enhanced performance at posttest. While multi-modal training yielded performance improvements only in trained tasks, physical training was associated with improvements in untrained working memory updating and immediate recall tasks, suggesting transfer effects to short-term and working memory functioning. In summary, the results demonstrate the importance of adaptive difficulty settings for short-term physical training interventions, at least for the enhancement of working memory.

## 1. Introduction

Physical activity does not only improve physical fitness, but also has the potential to enhance mental abilities and even protect against the age-related decline of cognitive and motor functions [1,2,3]. Endurance training has been shown to improve executive cognitive functions, possibly via the enhancement of angiogenesis and synaptic plasticity by brain-derived neurotropic factor (BDNF), a protein associated with neuron growth (e.g., [1,4,5,6,7]. In addition, interactions between improved cognition and cerebrovascular regulation as a result of aerobic exercise have been demonstrated [8]. Similarly, positive transfer effects of strength training have been reported for, among others, memory and attention functions (e.g., [9,10]). Resistance exercises have also been shown to evoke functional changes, especially in frontal brain areas (for review, [11]). Yet, while the close association between physical fitness and cognitive performance is, in principle, beyond question (e.g., [4]), the effectiveness of physical training on cognitive functions is not unanimously supported by empirical evidence. For example, a meta-analysis revealed no firm evidence that aerobic physical activities have any positive effects on the cognition of healthy older adults [12]. Moreover, there is still no consensus on the “best” way to improve cognitive functions with respect to physical exercise type (e.g., resistance, endurance, or coordination training), training intensity, and training duration. 

There is increasing evidence, however, that multi-modal training interventions are most effective for improving cognitive functions (for meta-analyses and reviews on the effects of combined cognitive and physical interventions, see [13,14]). The multi-modal training regimen usually targets motor and/or cognitive functions simultaneously. As such, multi-modal physical training interventions can include resistance, balance, aerobic, and stretching exercises, all of which have shown beneficial effects on cognitive performance [15,16]. Similarly, dual-task training, which calls for sensorimotor activity involving both physical and cognitive elements, has a positive influence not only on motor skills, but also on processing speed, attention, executive control, and mental rotation [17,18,19,20,21,22]. In a meta-analysis of 25 studies, Kelly and colleagues [23] examined the effects of a wide variety of different physical training regimens on cognitive functioning in seniors. They found that training interventions as different as resistance training and Tai Chi led to improvements in attentional functions and processing speed. Crucially, they also observed that combined training interventions led to greater effects than, for example, pure endurance training. Another meta-analysis of 52 studies demonstrated positive influences on the aging brain and cognition from both cognitive and physical training, while the benefits of combined cognitive and physical training have not yet been definitively proven [24].

A second important determinant for the success of a training regime appears to be its difficulty level. There is increasing evidence that training in which the task demands of the training exercises are dynamically adjusted to an individual’s current level of performance are especially suited to promoting training success. Such adaptive difficulty settings produce a prolonged mismatch between the individual’s level of performance and task demands. This mismatch is thought to be required to induce cognitive and physical plasticity and, with it, improve performance in trained as well as untrained tasks [25]. In keeping with this, larger transfer effects have been found with adaptive cognitive training than with non-adaptive cognitive training with consistently low task demands (e.g., [26,27]). In addition, successfully attaining a higher difficulty level in adaptive cognitive tasks has been shown to cause structural changes in the brain and promote motor learning [28]. The beneficial aspects of multi-modality and adaptive difficulty are both part of the Adaptive Capacity Model (ACM), an evolutionary–neuroscience model that links exercise, cognition, and brain capacity [29], assuming that the combination of physically and cognitively demanding exercises is beneficial to brain structure and function, as well as the enhancement of neuroplasticity.

Regarding the most effective training duration, the evidence is mixed. While positive effects of life-long/long-term physical activity are well documented (e.g., [30,31]), evidence of the impact of short-term (i.e., several week-long) physical exercise programs on cognitive functions is still limited. Cognitive improvements have been found relatively consistently after six months of physical training [10,32]. For example, seniors who completed two weekly one-hour sessions of resistance training or combined strength and balance training for six months showed significantly improved performance in a battery of standardized neuropsychological tests [15,16]. In their meta-analysis on a variety of physical training regimens, Kelly and colleagues [13] observed that, at least in older adults, longer-term interventions (>6 months) can be even more effective than interventions lasting six months or less. On the other hand, Nouchi and colleagues [33,34] were able to demonstrate the significant positive effects of a very short-term (i.e., four-week-long) physical training program on various cognitive abilities (e.g., episodic memory, working memory, attention, and processing speed) in healthy older adults. Yet, despite these promising results in older adults, the empirical evidence of the influence of short-term resistance and multi-modal training on cognitive and motor function during adulthood (20–60 years) is still scarce due to a lack of studies on this age range.

The aim of the present study was to address this issue and to assess the effects of a short-term training intervention on cognitive functions in a group of adult participants in a working environment. We compared two types of twelve-week-long physical training regimes (30 min each, once per week) with respect to their effects on cognitive functioning: purely physical strength training (physical training group) and multi-modal training combining physical and motor–cognitive training elements (combined training group). Both types of training featured adaptive difficulty settings in which task demands were dynamically adjusted to participant performance levels. For the combined (multi-modal) training, a novel movement-based coordination and reaction training device, the “Agility Board” (coordination and reaction board), was employed to apply motor–cognitive dual tasks. Cognitive–visual tasks (e.g., addition or working memory tasks) were to be solved as quickly as possible via gross motor and coordination-challenging movements (e.g., jumping with both legs on specific response fields; Figure 1A). This type of training has several advantages over traditional purely cognitive or athletic programs: brevity, variety, ease of use, continuous performance feedback, and, most importantly, adaptation to the trainee’s current abilities (adaptivity). In addition, it increases training motivation due to its game-like characteristics. The physical training group and the combined training group were compared with an active control group, which received a non-adaptive and low-intensity version of the purely physical strength workout performed by the physical training group. In an exploratory analysis, we tested whether the experimental and control groups differed in their improvements in cognitive performance, particularly regarding processing speed and attention. Special emphasis was placed on cognitive transfer effects, that is, training-related performance gains in untrained cognitive tasks. We therefore measured pre- and post-training performance in a comprehensive cognitive test battery that went beyond the cognitive functions targeted in the training protocols of the Agility Board.

## 2. Materials and Methods

### 2.1. Participants

A total of 124 adults who reported to be in good physical health were recruited from the staff of a German car manufacturing company as part of an in-house information campaign. The study was carried out in accordance with the Declaration of Helsinki and was approved by the local Ethics Committee of the Leibniz Research Centre for Working Environment and Human Factors, Dortmund, Germany. The participants gave written informed consent and received no financial compensation. They were assigned to either the physical training group (*n* = 41), the combined training group (*n* = 42), or the active control group (*n* = 41), according to a matching procedure that took into account age, gender, and work activity (e.g., leadership position, duration of employment, working hours, and shift work). Subjects who did not complete the training program were excluded from the analysis. During the intervention period, there were 22 dropouts due to illness, pregnancy, or employer change, leaving 102 subjects for analysis. Overall, 83.34% of the combined training group, 80.49% of the physical training group, and 82.93% of the active control group successfully completed the program (in compliance with the criteria). Table 1 provides an overview of the demographic characteristics (age, sex, and educational level) as well as physical activity (assessed with the Freiburger Fragebogen für körperliche Aktivität, fFkA; [35]) and crystalline intelligence (assessed with the Multiple-Choice Vocabulary Intelligence Test, MWT; [36]) of the sample. Separate one-way analyses of variance (ANOVAs) with the factor Group (physical training, combined training, or active control) did not show significant pre-intervention differences between groups in age, *F*(2, 99) = 0.141; *p* = 0.869; η_p_^2^ = 0.003, the fFkA overall activity score, *F*(2, 96) = 2.182; *p* = 0.118; η_p_^2^ = 0.043, or the MWT score, *F*(2, 89) = 1.363; *p* = 0.261; η_p_^2^ = 0.030. According to separate Χ^2^-tests, the three groups were also comparable with respect to gender distribution, Χ^2^ = 1.712; *p* = 0.425, and educational level, Χ^2^ = 3.879; *p* = 0.868.

### 2.2. Pretests and Posttests

Cognitive functions were examined using a computer-based test battery (ALAcog©) with different tasks assessing basic cognitive functions, such as perceptual processing, alertness, working memory updating, short-term memory, inhibition, perceptual speed, and mental rotation. The duration of the individual tasks varied from less than a minute to a maximum of four minutes. The following seven tasks were employed: Perceptual processing (M3): Selective attention/perceptual processing was assessed with the M3 test. During this computer-based letter focus task, participants were instructed to press one of two response keys when the upper- or lowercase target letter “M” surrounded by three dots appeared onscreen. When the upper- or lowercase letter “W” or the upper- or lowercase letter “M” surrounded by less than three dots were presented, participants had to press the other response key. The participants performed the task under time pressure (90 s) and had to respond as quickly and as accurately as possible. The mean response accuracy (percentage of correct responses) and mean response time (RT) were analyzed.Alertness (0-back): Participants performed a 0-back-task on a sequence of letters successively presented on the computer screen. If the currently shown letter was an “X”, a response key had to be pressed. A total of 102 letters were presented at intervals of 1500 ms, including 23 targets. The participants had to respond as quickly and as accurately as possible. The mean response accuracy and mean RT were analyzed.Working Memory Updating (2-back): Participants performed a 2-back-task on a sequence of letters successively presented on the computer screen. They had to press the response key if the current letter matched the letter presented two positions back in the sequence. A total of 156 letters were presented at intervals of 1500 ms, including 31 targets. The subjects had to respond as quickly and as accurately as possible. The mean response accuracy and mean RT were analyzed.Short-Term Memory (Immediate and Delayed Recall): Participants were instructed to memorize a list of 10 words that was presented for one minute. They then had to recall as many of the words as possible immediately after presentation (immediate recall) and 30 min later (delayed recall). The participants were given a 1.5 min time window during which they had to type in the words they recalled from the list via a computer keyboard. The immediate recall score was defined as the sum of correctly recalled words immediately after the presentation of the list. The delayed recall score was defined as the sum of correctly recalled words 30 min later. Two different versions of the test featuring different word lists were used for pretest and posttest measurement. The mean recall scores were analyzed separately for immediate and delayed recall.Inhibition (Stroop): A computer-based Stroop task [37] with three subtests was used to assess inhibition. First, the participants were presented with 20 trials, each featuring one of four color words (red, green, yellow, or blue) in white font. Their task was to indicate the meaning of the words by pressing an assigned key on the keyboard. Second, the participants were shown 20 trials, each featuring either a red, green, yellow, or blue rectangle, and had to indicate its color by pressing the assigned key. Third, 120 trials followed, each featuring one of the four color words printed in colored font. Participants were asked to indicate the font color by pressing the respective key. Font color was congruent to the word meaning on half of the trials (e.g., the word “green” presented in green font) and incongruent to it on the other half (e.g., the word “blue” presented in yellow font). The response time was limited to 2500 ms per trial. The mean Stroop effects in accuracy and RT (performance differences between congruent and incongruent trials) were calculated and analyzed.Perceptual Speed (TMT): A computerized version of the Trail Making Test (TMT) A and B [38] was used to assess perceptual speed. In a first subtest (TMT-A), the participants were presented with a display featuring the numbers from 1 to 26 at random positions. They were instructed to click on these numbers in ascending order as quickly as possible. In a second subtest (TMT-B), the display contained the numbers from 1 to 13 and the letters from A to M. The participants had to alternate between clicking on the numbers and letters in ascending order as quickly as possible. The mean response accuracy and mean overall completion time were analyzed.Mental rotation (Rotate Mirror): In this test, the participants viewed 26 trials, each featuring a row of five identical complex geometric shapes rotated into distinct positions along the horizontal axis. One of the five shapes was additionally mirror-reversed. The participants had to click on this target shape as quickly as possible. The maximum response time window was 30 s per trial. The mean response accuracy and mean RT were analyzed.

In addition to these purely cognitive tasks, motor–cognitive performance in the dual-task condition of the combined training was measured by using six different cognitive tasks of the Agility Board (Table 2). The Agility Board consisted of a tablet computer and a touch-sensitive response board (Figure 1A). The users were presented with different cognitive tasks on the tablet and had to respond to these tasks as quickly as possible by tapping or jumping on one or more of the five response fields on the board.

For each Agility Board task, except for the three sequence memory tasks, the mean response accuracy and the overall number of responses completed within the time limit of each task were analyzed as an index of cognitive processing speed. For the three sequence memory tasks, the dependent variables were the mean response accuracy and the set size levels achieved by the end of the individual time limits. 

### 2.3. Training Interventions

Both experimental groups (physical training or combined training) and the active control group underwent a weekly health-oriented exercise program for twelve weeks (Figure 1B). All aspects of the training programs were closely supervised by certified professional instructors. For the experimental groups, physical training intensity was progressively adjusted for each participant based on the Individual Lifting Performance Method [39]. A training session lasted for 30–35 min and consisted of a warm-up period (4 min on a bicycle ergometer) and either *physical* (strength) training (25–30 min) or *combined* training, that is, 10–15 min of strength training and 15 min of dual-task training on the Agility Board (motor–cognitive training). 

The strength training (physical training group and combined training group) consisted of circle resistance training, which included six to seven exercises for muscle-strengthening focused on the main muscle groups of the body. One set (combined training) or two sets (strength training) of 18 to 22 repetitions of each exercise were performed (time under tension: 60 s). Thirty-second breaks after each set were provided to change the resistance machine. The dual-task program of the combined training group additionally included the six motor–cognitive dual-tasks on the Agility Board (Table 2). The non-adaptive active control group also performed two sets of the resistance exercises of strength training, but was not subject to any intensity increase during the training period. Instead, the participants in this group trained at a maximum of 50% of their individually possible weight load after the 20 repetition maximum (RM) test [40] and maintained these individually determined resistance weights throughout the training period. 

Prior to the person-specific planning of the training, an instruction session was held with an explanation of the equipment and the training principle, as well as the determination of the individual muscle strength (X-RM method). A subjective questioning of the load feeling also took place. In addition, the subject’s deviation from the given movement speed, which could be interpreted as the beginning of neuromuscular fatigue, was also taken into account. All data obtained were recorded on a personal chip card, which also functioned as a training control system. Compliance with the program, regardless of the group, was ensured by the fact that this always took place under controlled conditions with specialist personnel.

### 2.4. Procedure

The training took place on the company’s premises in a mobile fitness studio that was set up as part of the company’s internal occupational health management program. The fitness studio was integrated in a truck trailer and offered a total of six training places. The participants performed all components of the study during their work time. There were no specific instructions during the measurements (e.g., regarding lifestyle and diet) due to the different operational conditions, such as different shift times and testing during working hours. Before the start of the program, it was ensured that all participants had neither health impairments nor serious illnesses. Pretest measurements were taken between one and two weeks prior to the start of the training intervention, and posttest measurements were taken five to ten days after the last training session.

### 2.5. Data Analyses

All analyses included only participants (*n* = 102) who provided complete datasets for the respective analyzed task. In order to test for group differences at baseline, pretest data of each task were analyzed in separate one-way ANOVAs with the between-subject factor Group (physical training, combined training, or active control). Training gains and transfer effects were analyzed by submitting the dependent variables described above to separate repeated-measures 3 × 2 ANOVAs for each task, with the between-subject factor Group (physical training, combined training, or active control) and the within-subject factor Test Time (pretest or posttest). For significant Group by Time interactions, we further examined whether training-related benefits in one or both of the experimental groups differed from those in the active control group, and whether efficacy differed between the combined training and the physical training alone. To this end, the posttest minus pretest difference values of all measures were computed and submitted to contrast analyses, which contrasted the training gains related to physical training, combined training, and the active control procedure. The level of significance was set as *p* < 0.05. For all post hoc tests and contrast analyses, Bonferroni-corrected *p*-values are reported. Partial eta square (η_p_^2^) values are provided as estimators of effect size.

## 3. Results

Table 3 offers an overview of the performance of the three experimental groups in all Agility Board and cognitive tasks at pretest and posttest. 

### 3.1. Pretest (Baseline) Performance 

The baseline analysis revealed no significant group differences at baseline (all *Fs* < 2.616, *ps* > 0.08, η_p_^2^ < 0.051), indicating that pretest performance was equivalent between the three groups regarding all dependent variables.

### 3.2. Agility Board-Related Training Gains and Transfer Effects

#### 3.2.1. Reaction Duo

Response accuracy was improved at posttest compared to pretest, resulting in a significant main effect of Time *F*(1, 96) = 10.2, *p* = 0.002, η_p_^2^ = 0.1. Neither the main effect of Group nor the interaction were significant (both *ps* > 0.68). Regarding the number of responses completed within the time limit of the task, there were significant main effects of Time, *F*(1, 96) = 0.37.68, *p* < 0.001, η_p_^2^ = 0.28, and Group, *F*(2, 96) = 6.1, *p* = 0.003, η_p_^2^ = 0.11, and a significant Time × Group interaction, *F*(2, 96) = 27.87, *p* < 0.001, η_p_^2^ = 0.37. Bonferroni-corrected post hoc tests showed performance gains from pretest to posttest only in the combined training group (*p* < 0.001), but not in the other groups (both *ps* > 0.55). Accordingly, training gains in the combined group were larger than those in the physical group and the active control group (both *ps* < 0.001).

#### 3.2.2. Forward Sequence Memory

The accuracy data showed significant main effects of Time, *F*(1, 96) = 23.84, *p* < 0.001, η_p_^2^ = 0.2, and Group, *F*(2, 96) = 7.86, *p* < 0.001, η_p_^2^ = 0.14, and a significant Time × Group interaction, *F*(2, 96) = 9.66, *p* < 0.001, η_p_^2^ = 0.17. Bonferroni-corrected post hoc tests showed increased accuracy at posttest compared to pretest only for the combined training group (*p* < 0.001), but not for the two other groups (both *ps* > 0.14). There were larger training gains in the combined group than in the physical group (*p* < 0.001) and the active control group (*p* = 0.002). Regarding the set size levels, we found significant main effects of Time, *F*(1, 96) = 28.31, *p* < 0.001, η_p_^2^ = 0.23, and Group, *F*(2, 96) = 8.08, *p* < 0.001, η_p_^2^ = 0.14, as well as a significant Time × Group interaction, *F*(2, 96) = 12.82, *p* < 0.001, η_p_^2^ = 0.21. Bonferroni-corrected post hoc tests showed performance improvements at posttest for both the physical training group (*p* = 0.005) and the combined training group (*p* < 0.001), but not for the active control group (*p* = 0.69). Again, training gains in the combined group were larger than those in the physical group (*p* = 0.011) and the active control group (*p* < 0.001). In addition, gains were also larger in the physical group than in the active control group (*p* = 0.035).

#### 3.2.3. Backward Sequence Memory

For response accuracy, there were significant main effects of Time, *F*(1, 96) = 22.5, *p* < 0.001, η_p_^2^ = 0.19, and Group, *F*(2, 96) = 5.82, *p* = 0.004, η_p_^2^ = 0.11, and a significant Time × Group interaction, *F*(2, 96) = 9.26, *p* < 0.001, η_p_^2^ = 0.16. Increases in accuracy from pretest to posttest were only significant for the combined training group (*p* < 0.001), but not for the two other groups (both *ps >* 0.08). Training gains were larger in the combined group than in the physical (*p* = *0*.004) and active control groups (*p* < 0.001). The analyses of the set size levels yielded significant main effects of Time, *F*(1, 96) = 21.62, *p* < 0.001, η_p_^2^ = 0.18, and Group, *F*(2, 96) = 5.76, *p* = 0.004, η_p_^2^ = 0.11, as well as a significant Time × Group interaction, *F*(2, 96) = 8.56, *p* < 0.001, η_p_^2^ = 0.15. Again, Bonferroni-corrected post hoc tests showed level improvements from pretest to posttest only for the combined training group (*p* < 0.001; both other *ps* > 0.23), and training gains were larger in the combined group than in the physical (*p* = 0.001) and active control groups (*p* < 0.001).

#### 3.2.4. Math Duo

The response accuracy data yielded significant main effects of Time, *F*(1, 96) = 4.83, *p* = 0.030, η_p_^2^ = 0.05, and Group, *F*(2, 96) = 4.84, *p* = 0.010, η_p_^2^ = 0.09, but no significant interaction between the two factors (*p* = 0.096). Response accuracy was higher at posttest compared to pretest and in the combined training group compared to the active control group (*p* = 0.003; both other comparisons, *ps* > 0.31). For response quantity, there were significant main effects of Time, *F*(1, 96) = 43.8, *p* < 0.001, η_p_^2^ = 0.31, and Group, *F*(2, 96) = 10.09, *p* < 0.001, η_p_^2^ = 0.17, as well as a significant Time × Group interaction, *F*(2, 96) = 27.74, *p* < 0.001, η_p_^2^ = 0.37. Bonferroni-corrected post hoc tests indicated performance gains from pretest to posttest only in the combined training group (*p* < 0.001; both other *ps* > 0.22); gains were larger in the combined group than in the two other groups (both *ps* < 0.001).

#### 3.2.5. Rotation

Response accuracy showed a significant main effect of Time, *F*(1, 96) = 18.62, *p* < 0.001, η_p_^2^ = 0.16, due to higher accuracy at posttest compared to pretest. The main effect of Group, *F*(2, 96) = 3.55, *p* = 0.033, η_p_^2^ = 0.07, was also significant due to the higher accuracy in the combined training group than in the active control group (*p* = 0.029; both other comparisons, *ps* > 0.35). The Time × Group interaction did not reach significance (*p* = 0.31). The analyses of response quantity yielded significant main effects of Time, *F*(1, 96) = 236.94, *p* < 0.001, η_p_^2^ = 0.71, and Group, *F*(2, 96) = 23.37, *p* < 0.001, η_p_^2^ = 0.33, and a significant Time × Group interaction, *F*(2, 96) = 82.88, *p* < 0.001, η_p_^2^ = 0.63. Bonferroni-corrected post hoc tests indicated performance gains from pretest to posttest in both training groups (both *ps* < 0.001) and in the active control group (*p* = 0.013). Gains were larger in the combined group than in the two other groups (both *ps* < 0.001).

#### 3.2.6. Forward Sequence Memory Duo 

The accuracy data yielded significant main effects of Time, *F*(1, 96) = 6.7, *p* = 0.011, η_p_^2^ = 0.07, and Group, *F*(2, 96) = 6.37, *p* = 0.003, η_p_^2^ = 0.12, but no significant Time × Group interaction (*p* = 0.081). The response accuracy was higher at posttest compared to pretest, and higher in the combined training group than in the active control group (*p* = 0.002), while the combined and physical training groups, as well as physical training and active control groups, did not differ from each other (both *ps* > 0.1). The analyses of the set size levels showed significant main effects of Time, *F*(1, 96) = 9.75, *p* = 0.002, η_p_^2^ = 0.09, and Group, *F*(2, 96) = 4.58, *p* = 0.013, η_p_^2^ = 0.09, indicating increased levels at posttest compared to pretest and in the combined training group relative to the active control group (*p* = 0.010; both other *ps* > 0.38). The Time × Group interaction just missed significance, *F*(2, 96) = 2.84, *p* = 0.063, η_p_^2^ = 0.06. Bonferroni-corrected post hoc tests showed increased levels at posttest compared with pretest only in the combined training group (*p* < 0.001; both other *ps* > 0.3), while training gains did not differ significantly between groups.

#### 3.2.7. Summary—Agility Board-Related Training Gains and Transfer Effects

Regarding response accuracy, four of the six Agility Board tasks (Reaction Duo, Rotation, Math Duo, and Forward Sequence Memory Duo) showed retest effects (i.e., equivalent performance gains from pretest to posttest in all groups). For the Forward and Backward Sequence Memory tasks, we found accuracy improvements, which were limited to the combined training group. Regarding the task processing speed, three of the six Agility Board tasks (Reaction Duo, Math Duo, and Backward Sequence Memory) showed such training-related benefits, which were limited to the combined training. The Forward Sequence Memory Duo task also showed small, but non-significant, speed benefits at posttest only after combined training. In the Rotation task, all groups showed retest effects, yet the performance gains of the combined training group significantly exceeded those of the other groups. For the Forward Sequence Memory task, the set size levels were increased at posttest for both training groups, but not for the active control group. Physical training was thus associated with a transfer effect to this (for the strength training group) untrained task.

### 3.3. Cognitive Transfer Effects

#### 3.3.1. Perceptual Processing (M3)

Regarding response accuracy, neither the main effects of Time and Group, nor the Time × Group interaction, reached significance (all *ps* > 0.3). Across all groups, the mean RTs were shorter at posttest compared to pretest, as reflected by the significant main effect of Time, *F*(1, 99) = 5.53, *p* = 0.021, η_p_^2^ = 0.05. The main effect of Group and the Time × Group interaction did not reach significance (both *ps* > 0.3).

#### 3.3.2. Alertness (0-Back)

No significant effects emerged for either the response accuracy (both *ps* > 0.06) or reaction time data (all *ps* > 0.56).

#### 3.3.3. Working Memory Updating (2-Back)

The accuracy data yielded no significant main effects of Time or Group (both *ps* > 0.15), but a significant Time × Group interaction, *F*(2, 99) = 3.37, *p* = 0.038, η_p_^2^ = 0.06. Bonferroni-corrected post hoc tests indicated that accuracy was increased at posttest compared to pretest only for the physical training group (*p* = 0.045; both other *ps* > 0.08). Moreover, the contrast analyses indicated larger training gains in the physical group than in the active control group (*p* = 0.032), while the combined group and active control group did not differ significantly (p = 0.052). The mean RT yielded a significant main effect of Time, *F*(1, 99) = 24.42, *p* < 0.001, η_p_^2^ = 2, due to the shorter overall RT at posttest compared to pretest, but no other significant effects (both *ps* > 0.88).

#### 3.3.4. Short-Term Memory (Immediate and Delayed Recall)

The mean response accuracy of immediate recall yielded no significant main effects of Time and Group (both *ps* > 0.098), but a significant Time × Group interaction, *F*(2, 99) = 5.17, *p* = 0.007, η_p_^2^ = 0.1. Bonferroni-corrected post hoc tests showed increased accuracy at posttest compared to pretest only for the physical training group (*p* < 0.001; both other *ps* > 0.61). Training gains were larger in the physical group than in the active control group (*p* = 0.007) and in the combined group (*p* = 0.013). The mean response accuracy of delayed recall showed a significant main effect of Time, *F*(1, 99) = 9.95, *p* = 0.002, η_p_^2^ = 0.09, which was due to the higher overall accuracy at posttest compared to pretest, but no other significant effects (all *ps* = 0.38).

#### 3.3.5. Inhibition (Stroop)

A significant main effect of Time on accuracy, *F*(1, 99) = 13.77, *p* < 0.001, η_p_^2^ = 0.12, resulted from a reduction in the Stroop effect from pretest to posttest. None of the other effects reached significance (all *ps* > 0.08). The analyses of the RT Stroop effect also yielded a significant main effect of Time, *F*(1, 99) = 14.13, *p* < 0.001, η_p_^2^ = 0.13, due to a reduction from pretest to posttest. The other effects were not significant (all *ps* > 0.49).

#### 3.3.6. Perceptual Speed (TMT)

The accuracy of TMT-A yielded no significant effects (all *ps* > 0.32). The mean completion time was shorter at posttest compared to pretest, resulting in a significant main effect of Time *F*(1, 99) = 7.99, *p* = 0.006, η_p_^2^ = 0.08. No other effects reached significance (both *ps* > 0.15). The analyses of accuracy and completion time of TMT-B yielded no significant effects (all *ps* > 0.06).

#### 3.3.7. Mental Rotation (Rotate Mirror)

No significant effect occurred for accuracy data (all *ps* > 0.06). The mean RTs were shorter at posttest compared to pretest, as indicated by the main effect of Time, *F*(1, 99) = 13.1, *p* < 0.001, η_p_^2^ = 0.12. No other effects reached significance in either accuracy (both *ps* > 0.33) or RT (both *ps* > 0.32).

#### 3.3.8. Summary—Cognitive Transfer Effects

For all groups, retest effects were found in the tasks measuring Perceptual Processing, Mental Rotation, and Delayed Recall, as well as the Stroop and TMT-A tasks. Transfer effects emerged only for the physical training group regarding response accuracy in the untrained Working Memory and Immediate Recall tasks.

## 4. Discussion

The aim of the present study was to examine the benefits of two adaptive short-term training interventions on cognitive functions. The effects of adaptive purely physical strength training and adaptive multi-modal combined physical and motor–cognitive training were contrasted with an active control condition consisting of a non-adaptive and low-intensity physical strength workout. The multi-modal training condition combined physical strength training with motor–cognitive tasks performed on the Agility Board, a novel movement-based coordination and reaction training device. Regarding the Agility Board tasks, significant performance improvements from pretest to posttest were mainly found in the combined training group. As the pre- and posttest Agility Board tasks were the same as those used during this group’s training, these improvements were most likely due to their training experience on the Agility Board. Agility Board transfer effects were scarce: physical training improved performance only in the Forward Sequence Memory task relative to the active control condition. Still, physical training also yielded transfer effects in the non-trained cognitive tasks, with performance improvements from pretest to posttest found in the Working Memory Updating and Immediate Recall tasks of the test battery we employed. Physical training generated no performance gains in measures of perceptual processing, perceptual speed, alertness, delayed memory recall, inhibition, or mental rotation, however. Moreover, the cognitive tasks showed no transfer effects whatsoever for the combined training group. Overall, transfer effects were thus limited to a positive effect of pure physical training on (motor–)cognitive tasks drawing on short-term and working memory processes. 

Improvements in untrained cognitive functions due to physical training are in line with a number of previous studies (e.g., [4]). The present findings add to this well-known effect of “mens sana in corpore sano”, as they show that only 30 min of weekly high-intensity adaptive physical exercise over twelve weeks was sufficient to improve short-term and working memory functions relative to the active control condition. Moderate physical training intensity, training unit lengths of more than half an hour, as well as training programs stretching over longer time periods have frequently been propagated as optimal (e.g., [1,41]). The present results run contrary to these assumptions and instead confirm observations made in more recent studies that examined similarly short intervention and/or training unit durations, as well as higher exercise intensity (e.g., [42]). For example, Nouchi and colleagues [33,34] demonstrated benefits to short-term and working memory after only twelve high-intensity 30 min (endurance- and strength-based) training sessions. The benefits of short, but at least moderate, physical training on cognitive functions are also supported by a meta-analysis of 39 studies with older participants [43]. Transfer effects related to the short-term physical training we applied were similarly limited to untrained cognitive tasks that draw on short-term and working memory functions, i.e., tasks measuring working memory updating and immediate, but not delayed, memory recall. In line with this, a meta-analysis of the effects of physical exercise on cognitive performance also showed training-related benefits, especially for tasks drawing on executive functions, such as selective attention and working memory [1].

In the present study, the physical training group performed the same physical strength workout as the active control group, but at an adaptively increasing intensity level. In contrast to a large number of previous training studies, we did not use a passive control group, but compared performance improvements in adaptively versus non-adaptively trained groups. A recent review on the effects of physically active video gaming showed that the choice of control group can have a substantial effect on the observed cognitive benefits [44]. Comparing training groups to active rather than passive control groups decreases the likelihood of finding significant effects and narrows the range of cognitive domains in which they are found. Passive control groups without any intervention considerably differed from training groups in their general activity level. Active control groups engaged in alternate interventions eliminated this discrepancy and thus yielded fewer and more specific observations of training-related benefits. Pinpointing specific training parameters underlying training success may still be fairly difficult, however, as training interventions and active control group interventions usually differ along several dimensions. In the present study, we addressed this issue by having the active control group perform a non-adaptive version of our physical strength training. As a result, we observed training-related benefits only in tasks drawing on short-term and working memory functions. Finally, it should be mentioned that a potential disadvantage of using an active control group is that it might be difficult to maintain a stable physiological state during the study period. To capture these effects, a comparison with a passive control group would be useful. However, an additional control group was not feasible in this study due to time and practicality constraints.

At least for transfer to the working memory domain, the present findings corroborate the importance of continuously adapting physical exercise demands to the current performance level, which is quite in line with previous findings on the impact of adaptive difficulty levels on both cognitive and physical training success (e.g., [25,26,27]). This key role of adaptive difficulty is also consistent with the Adaptive Capacity Model (ACM; [29]) and the assumption that a prolonged mismatch between the individual’s level of performance and task demands is required to alter cognitive performance in trained tasks and to promote transfer effects and cognitive plasticity. In particular, it is assumed that the mismatch between the current functional capacity of the system and the environmental demands has to be located within a current range of flexibility in order to trigger the enhancement of neuroplasticity, with task demands being neither too low nor too high [25]. 

The combined multi-modal training did not show transfer effects to any of the untrained cognitive tasks, but merely yielded performance gains in trained (Agility Board) tasks. The only transfer effect was a (non-significant) tendency for a better performance in the 2-back task in the multi-modal vs. the control group. The combined training primarily increased the task processing speed. For five of the six Agility Board tasks, we found speed benefits following combined training that significantly exceeded those in the other two groups. The Forward Sequence Memory Duo task also showed small speed benefits only after combined training. These did not reach significance, however, which was probably due to the high difficulty level of this task, as illustrated by the low accuracy rates across all groups and test times. Speed improvements across (almost) all tasks after combined training probably reflect training-related enhancements in the handling of the Agility Board, a novel device requiring some coordinative expertise, rather than benefits to specific motor–cognitive functions. 

Regarding task accuracy, Reaction Duo, Rotation, Math Duo, and Forward Sequence Memory Duo showed retest effects, that is equivalent performance gains from pretest to posttest in all groups. For the two medium-complexity tasks involving working memory processes, Forward and Backward Sequence Memory, accuracy improvements were, however, limited to the combined training group. Notably, practice-related accuracy benefits associated with combined motor–cognitive training thus only emerged in the working memory tasks, as did the transfer effects we observed after physical strength training.

Previous studies on cognitive training have reported contradictory results regarding transfer effects to untrained tasks. While transfer effects were limited or absent in some studies (e.g., [41,45,46,47,48]), others have found evidence of transfer (e.g., [27,49,50]). Particularly successful in terms of promoting transfer effects seem to be complex, multi-faceted, and adaptive intervention strategies that involve cross-domain cognitive components (e.g., [51,52,53]). Dual-task training can also be highly effective [54] and, according to some research groups, working memory plays a decisive role in this context (e.g., [26,54,55]). 

The present results do not support these assumptions regarding transfer effects and are more in line with a meta-analysis, suggesting only very minor transfer effects of combined physical and cognitive training on cognition [13]. Although the combined training we employed was complex, multi-faceted, adaptive, and featured dual tasks involving working memory, it yielded only a slight (non-significant) transfer effect on the untrained Working Memory Updating (2-Back) task of the cognitive test battery. Moreover, the combined training did not yield greater performance benefits than physical training in any of the untrained cognitive tasks. One possible explanation for this may be the fact that the combined group only performed half the amount of strength training that the physical training group did (i.e., 10–15 min per session compared to 25–30 min per session). There is ample evidence to suggest that the amount and length of physical training performed modulates the magnitude of training-related cognitive benefits (e.g., [23]). It is possible that the 10–15 min of strength training performed by the combined group were insufficient to promote the transfer effects we observed for the physical training group after 25–30 min of strength training per session. Instead of further strength training, the combined group performed 10–15 min of motor–cognitive training on the Agility Board, which was also physically demanding and, at least for three out of six tasks, adaptive (cf. Table 2). This should have resulted in a continuously increasing cognitive and physical task load during training, which, however, yielded fewer transfer effects than the increasing physical task load associated with continuous strength training. In this context, a study by Küper and Karbach [56] is of interest, which compared the benefits of a single-task working memory training intervention to those of a more complex dual-task version of the training. Both training regimens were very brief and involved only five training sessions. As in the present study, the authors observed more extensive cognitive transfer effects for the simpler single-task version of their training. They argued that the complex cognitive learning processes involved in dual-task cognitive training may require a sufficient training length in order to promote transfer effects and that, in the short-term, simpler single-task training may be more effective. The present findings lend support to this assumption and show that it may not only apply to purely cognitive dual tasks, but also to motor–cognitive dual tasks. 

This raises the question as to what extent the performance improvements found in the combined group can be traced back to the use of the Agility Board with its specific motor–cognitive training features as opposed to the physical strength workout, which each made up half of this group’s training. Previous research suggests that it may be the motor-coordinative aspects of physical training (as amply provided by the Agility Board) that lead to cognitive improvements. Voelcker-Rehage et al. [31], for example, confirmed benefits to cognitive functions (including reactivity or mental rotation) after training components involving high demands on coordination (for similar results, see [17,18,19,20]). A future study in which the effects of Agility Board training alone, without any additional physical training, are tested may answer this question, especially if training durations are kept constant in both training conditions. Another limitation of the present study may be the training duration of the motor–cognitive training in the combined group. In order to keep the total duration of the intervention constant across the three groups, the combined group received only 15 min of training on the Agility Board. As with the physical component of the combined training (see above), this may have been too short to induce significant transfer effects. Future studies should thus systematically extend the duration of Agility Board training sessions in order to examine to what extent training length modulates the magnitude of training gains and transfer effects in the motor–cognitive domain. 

Finally, it should be noted that the entire training program is to be seen in a workplace context. All subjects were employed in a medium-sized company in the automotive sector and performed the pretests and posttests, as well as the training itself, during their working hours. For this purpose, a mobile fitness studio was used, which was integrated in a truck trailer. The present intervention approach could thus indicate a way to improve the physical and mental fitness of employees through short-term training programs at the workplace and thus counteract negative effects, especially those of occupations with predominantly sedentary work. Workplace health programs in general and specific training interventions in particular have the potential to reduce health risks and improve the quality of life of workers. There is evidence that cognitive functioning depends, at least in part, on the type of work performed by an individual [57,58]. However, research has also shown that the training of employees can have beneficial effects on their cognitive abilities and mental health, which can counteract the deleterious effects of adverse working conditions [59,60]. The present results indicate that only 30 min of physical training once per week at the workplace may not only have beneficial effects on physical performance, but also on cognitive performance. In the future, mobile training centers like that employed here could be a promising way for occupational health management to keep employees physically and mentally healthy.

## 5. Conclusions

In summary, the results show differences in the efficiency of the training interventions compared to the control group: short-term physical training consisting of short training unit durations in combination with high exercise intensity appeared to be highly effective for improving even untrained cognitive functions, with transfer effects seen mainly in working memory functions. The results demonstrate the importance of adaptive training difficulty, in line with the Adaptive Capacity Model, assuming the necessity of a prolonged mismatch between the individual performance level and current task demands. They also show the advantages of an active (but non-adaptively trained) control group over a passive control group in the evaluation of the effectiveness of training interventions. Finally, multi-modal training was not superior to purely physical training in terms of transfer to cognitive functions, while the possible influences of training duration require further investigation.

## Figures and Tables

**Figure 1 ijerph-19-07506-f001:**
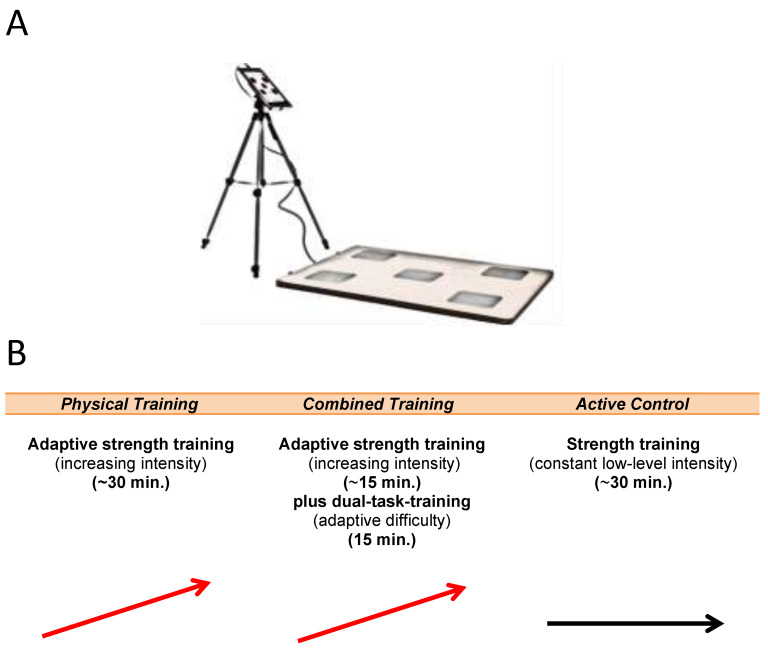
(**A**) Agility Board. This motor–cognitive training device consists of a tablet computer and a touch-sensitive response board. Users are presented with different cognitive tasks on the tablet and have to respond to these tasks by tapping or jumping on one or more of the 5 response fields on the board as quickly as possible. (**B**) Overview of the two training groups and the active control group. Adaptive strength training is symbolized by a red arrow, constant training by a black arrow.

**Table 1 ijerph-19-07506-t001:** Demographic characteristics, physical activity, and crystalline intelligence of the participant sample as a function of group (physical training, combined training, or active control).

		Physical Training		Combined Training		Active Control	
N		33		35		34	
Sex Distribution	Males (N)	28		30		32	
	Females (N)	5		5		2	
Highest Educational Degree (Years of Education)	12 Years (N)	2		2		4	
	11 Years (N)	7		9		7	
	10 Years (N)	14		13		11	
	9 Years (N)	10		10		12	
	Other	0		1		0	
Age Range (Years)		20–61		20–52		21–60	
		M	(SE)	M	(SE)	M	(SE)
Age (Years)		36.1	(1.8)	37.1	(1.5)	37.3	(1.5)
fFkA Overall Activity Score		48.5	(7)	41.5	(6.7)	30.5	(4.1)
MWT score		28	(0.8)	29	(0.5)	27.4	(0.8)

Note: M indicates the group mean and SE indicates the standard error of the mean.

**Table 2 ijerph-19-07506-t002:** Measure of motor–cognitive skills using the Agility Board.

Test	Target Dimension	Task Description	Dependent Variables
Agility Board 1: Reaction Duo	Reaction and co-ordination	Two randomly selected response fields are highlighted on the tablet screen. The participants’ task is to jump onto these two fields with both feet simultaneously as quickly as possible. (Task duration: 60 s)	Accuracy and overall number of responses completed within time limit
Agility Board 2: Forward Sequence Memory	Working memory	Onscreen, the response fields are highlighted sequentially. The participants’ task is to memorize the sequence and tap the respective fields on the board with one foot in the correct (forward) order. The task’s difficulty is adaptable ^1^. (Task duration: 180 s)	Accuracy and set size level achieved
Agility Board 3: Backward Sequence Memory	Working memory	Onscreen, the response fields are highlighted sequentially. The participants’ task is to memorize the sequence and tap the respective fields on the board with one foot in reverse order. The task’s difficulty is adaptable ^1^. (Task duration: 180 s)	Accuracy and set size level achieved
Agility Board 4: Math Duo	Visual search and processing speed	Onscreen, an addition term is presented showing only the sum, but leaving the summands blank (e.g., _ + _ = 8). Below the addition term, the response fields of the board are depicted superimposed with numbers, which could potentially be the summands. The participants’ task is to jump onto the two response fields assigned to the numbers that yield the displayed sum. (Task duration: 60 s)	Accuracy and overall number of responses completed within time limit
Agility Board 5: Rotation	Mental rotation, processing speed, and inhibition	Onscreen, a target response field is highlighted, along with a direction arrow. The participants’ task is to mentally rotate this display in the direction indicated by the arrow and then tap the field corresponding to the target field on the board. (Task duration: 60 s)	Accuracy and overall number of responses completed within time limit
Agility Board 6: Forward Sequence Memory Duo	Working memory	Onscreen, pairs of response fields are highlighted sequentially. The participants’ task is to memorize the sequence and jump on the respective field pairs on the board with both feet in the correct (forward) order. The task’s difficulty is adaptable ^1^. (Task duration: 180 s)	Accuracy and set size level achieved

Note: Task difficulty in ^1^ is adapted to individual performance by lengthening/shortening the sequence after three correct/incorrect responses in succession, respectively.

**Table 3 ijerph-19-07506-t003:** Means (SD) of task performance at pretest and posttest as a function of group (physical training, combined training, or active control) for the Agility Board tasks and cognitive tasks, results of ANOVAs with the main effects of Time and Group and Time × Group interactions, and results of contrast analyses of the training gains of the three groups.

		Physical Training		Combined Training		Active Control					Contrast Analysis		
		Pretest	Posttest	Pretest	Posttest	Pretest	Posttest	Time	Group	IA	PT > AC	CT > AC	CT > PT
Agility Board Tasks													
Reaction Duo	Acc	87.3 (1.2)	90.2 (1.4)	88.2 (1.1)	91.4 (1.3)	87.3 (1.2)	90.3 (1.4)	0.002					
	Resp	56.9 (1.5)	57.7 (2)	**55.1 (1.5)**	**71.1 (2)**	55.2 (1.6)	56.2 (2.1)	<0.001	0.003	<0.001		<0.001	<0.001
Rotation	Acc	89.2 (1.8)	91.5 (1.2)	90.1 (1.7)	96.1 (1.1)	86.1 (1.8)	91 (1.2)	<0.001	0.033				
	Resp	**42.1 (1.5)**	**48.3 (1.6)**	**43.2 (1.4)**	**68 (1.6)**	**41.2 (1.5)**	**44.6 (1.7)**	<0.001	<0.001	<0.001		<0.001	<0.001
Math Duo	Acc	86.7 (1.8)	88.3 (1.9)	87.4 (1.7)	93.8 (1.8)	84 (1.8)	84.1 (2)	0.030	0.010	0.096			
	Resp	16.4 (0.7)	17.2 (0.8)	**16.7 (0.6)**	**23.3 (0.7)**	16.2 (0.7)	16.5 (0.8)	<0.001	<0.001	<0.001		<0.001	<0.001
Forward Sequence Memory	Acc	76.8 (1.4)	78.8 (1.3)	**77.2 (1.3)**	**85.8 (1.2)**	74.9 (1.4)	75.7 (1.3)	<0.001	<0.001	<0.001		<0.001	0.002
	Level	**4.7 (0.1)**	**5.2 (0.1)**	**4.9 (0.1)**	**5.9 (0.1)**	4.8 (0.1)	4.8 (0.1)	<0.001	<0.001	<0.001	0.035	<0.001	0.011
Backward Sequence Memory	Acc	75.2 (1.2)	77.5 (1.4)	**75 (1.1)**	**82.9 (1.3)**	73.5 (1.2)	73.9 (1.4)	<0.001	0.004	<0.001		<0.001	0.004
	Level	4.7 (0.1)	4.9 (0.2)	**4.7 (0.1)**	**5.6 (0.1)**	4.5 (0.2)	4.6 (0.2)	<0.001	0.004	<0.001		<0.001	0.001
Forward Sequence Memory Duo	Acc	65.9 (1.2)	66.3 (1.1)	65.6 (1.2)	70.1 (1.1)	62.5 (1.3)	63.8 (1.2)	0.011	0.003	0.081			
	Level	3.4 (0.1)	3.6 (0.1)	**3.5 (0.1)**	**4.0 (0.1)**	3.2 (0.1)	3.3 (0.1)	0.002	0.013	0.063		0.087	0.051
Cognitive Tasks													
M3: Perceptual Processing	Acc	97.5 (1.8)	95.9 (1.1)	94.2 (1.8)	95.1 (1)	94.9 (1.8)	94.4 (1.8)						
	RT	660.5 (23.2)	617.9 (34.3)	652.9 (22.5)	577.4 (33.4)	668.7 (22.9)	660.9 (33.8)	0.021					
Mental Rotation	Acc	77.6 (3.4)	77.4 (3.4)	79.8 (3.3)	84.7 (3.3)	74.9 (3.3)	78.5 (3.3)	0.060					
	RT	8641.6 (546.6)	7570.3 (493)	7940.7 (530.8)	7613.3 (478.7)	8888.9 (538.5)	8090.6 (485.6)	<0.001					
Stroop	Acc-D	7.3 (1.9)	2.7 (1.1)	5.7 (1.8)	3.5 (1.1)	11.2 (1.9)	4.7 (1.1)	<0.001					
	RT-D	158.5 (15.7)	133.2 (16.6)	155.1 (15.3)	107.7 (16.1)	161.7 (15.5)	136.3 (16.4)	<0.001					
2-back: Working Memory Updating	Acc	**79.8 (3)**	**85.1 (3.7)**	82 (2.9)	86.5 (3.6)	81.1 (3)	77.8 (3.6)			0.038	0.032	0.052	
	RT	462.9 (18.5)	424.1 (16.6)	455.1 (18)	414 (16.19	461.3 (18.2)	428.9 (16.3)	<0.001					
0-back: Alertness	Acc	96.8 (2.3)	100 (0.1)	97.1 (2.2)	99.9 (0.1)	98.2 (2.2)	99.6 (0.1)	0.062					
	RT	299.7 (10.7)	297.2 (7.8)	310 (10.4)	299.7 (7.6)	306.3 (10.5)	309.7 (7.7)						
TMT-A	Acc	97.2 (0.8)	97.1 (1)	96.6 (0.8)	96.1 (0.9)	94.9 (0.8)	96.8 (0.9)						
	Time	44,927.9 (2043.2)	43,797.9 (2197.9)	46,592.7 (1984)	41,613.3 (2134.2)	50,835.2 (2012.9)	46,244 (2165.4)	0.006					
TMT-B	Acc	95.2 (2)	95 (1.4)	91.7 (1.9)	93.5 (1.3)	92.9 (1.9)	94.4 (1.4)						
	Time	54,005.8 (4760.2)	56,449.1 (4203.2)	50,686.1 (4622.2)	55,600.7 (4081.3)	42,115.4 (4689.7)	45,767.5 (4140.9)		0.055				
Immediate Recall	Acc	**74.5 (3.2)**	**84.8 (3)**	80.6 (3.1)	80 (2.9)	76.2 (3.1)	74.7 (3)	0.099		0.007	0.007		0.013
Delayed Recall	Acc	67.7 (3.6)	77 (3.4)	69.7 (3.5)	76.3 (3.3)	66.8 (3.6)	69.4 (3.3)	0.002					

Note: Acc indicates the percentage of accurate responses, Resp indicates the number of responses within the time limit of the task in ms, Level indicates the set size level achieved at the end of the task, RT indicates reaction times, Acc-D indicates the difference in accuracy between congruent and incongruent trials, RT-D indicates the difference in reaction times between congruent and incongruent trials, and Time indicates the overall time needed to complete the task in ms. Time × Group interactions leading to significant within-group pre-/post-training differences are printed in bold. For the contrast analyses, PT > AC indicates larger training gains in physical than active control groups, CT > AC indicates larger gains in combined than active control groups, and CT > PT indicates larger gains in combined than physical training groups. All *p*-values < 0.1 are specified.

## Data Availability

The data presented in this study are available on request from the corresponding author. The data are not publicly available due to legal and ethical reasons.

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
