# Peer review of "A Comparison of the Effects of Short-Term Physical and Combined Multi-Modal Training on Cognitive Functions"

_ijerph, 2022, doi:10.3390/ijerph19127506_

Round 1

Reviewer 1 Report

The subject of the study and also the type of experiments conducted are certainly worthy of publication. I also strongly recommend the publication.
However, the evaluation and interpretation are flawed. 
First, nonspecific hypotheses are formulated. This is a problem because without a predefinition of power (or type II error), the interpretation of the results can only be very vague and basically no hypothesis can be validly tested.
Moreover, a large number of dependent variables are tested without taking into account the problem of multiple endpoints (see familywise error rate (FWER) and data snooping).

In addition:
It is mandatory that contrasts be defined before the study is conducted and not afterwards. In this study, only the type I error is corrected based on multiple comparisons. This is more or less wrong when testing a hypothesis.

My suggestion: 
Don't mention hypothesis testing. Instead, describe an exploratory study with the goal of finding significant associations and possibly deriving hypotheses that will be tested in a separate study (in another paper). This approach would be serious and the study would lose the bitter taste of data snooping (whether intentional or unintentional). 

Author Response

The subject of the study and also the type of experiments conducted are certainly worthy of publication. I also strongly recommend the publication.
However, the evaluation and interpretation are flawed. 
First, nonspecific hypotheses are formulated. This is a problem because without a predefinition of power (or type II error), the interpretation of the results can only be very vague and basically no hypothesis can be validly tested. Moreover, a large number of dependent variables are tested without taking into account the problem of multiple endpoints (see familywise error rate (FWER) and data snooping).

In addition:
It is mandatory that contrasts be defined before the study is conducted and not afterwards. In this study, only the type I error is corrected based on multiple comparisons. This is more or less wrong when testing a hypothesis.

My suggestion: 
Don't mention hypothesis testing. Instead, describe an exploratory study with the goal of finding significant associations and possibly deriving hypotheses that will be tested in a separate study (in another paper). This approach would be serious and the study would lose the bitter taste of data snooping (whether intentional or unintentional). 

Response: We thank the reviewer for the constructive suggestion and have rewritten the manuscript accordingly. Instead of a hypothesis-driven approach, we now emphasize the exploratory nature of the analyses in the introduction and also in the discussion.

Reviewer 2 Report

The purpose of this manuscript was to investigate the effects of adaptive and multi-modal short-term training interventions on cognitive training gains and transfer effects. The findings show that 12-week physical training consisting of short training unit durations in combination with high exercise intensity appears highly effective for improving even untrained cognitive functions, with transfer effects seen mainly in working memory functions. This study suggests the importance of adaptive difficulty settings for short-term physical training interventions, at least for the enhancement of working memory. This study presents interesting data and novel findings to the literature. Overall, the manuscript is scientifically sound and well written. Yet, I have some concerns.

1.     To review the manuscript well, numbering the pages and lines is needed.

2.     There is a need to explain the content of ‘Agility Board’ in the Introduction.

3.     Are there any requirements (i.e. fasting, diet, free of coffee) for pre-and post-measures after the intervention? How long is the time interval between the last training session and cognitive function measure?

4.     It is not easy to maintain a stable physiological state during the study period. Thus, a no-exercise or passive control group is the main limitation of this manuscript, despite the authors stating some justifications. Adding this limitation to the manuscript sounds more reasonable.

5.     Were there any indicators (ratings of perceived exertions, heart rate, and/or training impulse) to reveal that the participants completed the exercise trials according to the scheduled protocols?

6.     Figure B needs to be placed on the same page.

7.     It is wired to include a reference (Raichlen and Alexander, 2017) in Conclusion. 

Author Response

  1. To review the manuscript well, numbering the pages and lines is needed.

Response: Page and line numbers were added.

  1. There is a need to explain the content of ‘Agility Board’ in the Introduction.

Response: A short description of the Agility Board and its advantages over conventional training equipment was added.

  1. Are there any requirements (i.e. fasting, diet, free of coffee) for pre-and post-measures after the intervention? How long is the time interval between the last training session and cognitive function measure?

Response: It is now described that it was not possible to make explicit demands on the pre- and posttests with regard to lifestyle or behavior due to the different operational conditions (e.g. different shift times and testing during working hours). Posttest measurements were taken five to ten days after the last training session.

  1. It is not easy to maintain a stable physiological state during the study period. Thus, a no-exercise or passive control group is the main limitation of this manuscript, despite the authors stating some justifications. Adding this limitation to the manuscript sounds more reasonable.

Response: This potential limitation was mentioned as a limiting factor in the discussion.

  1. Were there any indicators (ratings of perceived exertions, heart rate, and/or training impulse) to reveal that the participants completed the exercise trials according to the scheduled protocols?

Response: Prior to the person-specific planning of the training, an instruction session was held with an explanation of the equipment and the training principle as well as the determination of the individual muscle strength (X-RM method). A subjective questioning of the load feeling also took place. In addition, the subject’s deviation from the given movement speed, which can be interpreted as the beginning of neuromuscular fatigue, was also taken into account. All data obtained were recorded on a personal chip card, which also functioned as a training control system.

  1. Figure B needs to be placed on the same page.

Response: The placement is certainly a matter of layout and will be shown correct in the final version.

  1. It is wired to include a reference (Raichlen and Alexander, 2017) in Conclusion.

Response: corrected 

Reviewer 3 Report

The study by Kardys and colleagues tests the effects different interventions programmes (one multi-modal and one strength-based) on cognition in a sample of adults. While the question is not novel, it remains extremely important and has significant implications for the field of improving and maintaining cognitive function across the lifespan. Importantly, the authors found evidence of cognitive transfer effects after 12 weeks of physical training. I have a few suggested improvements for the authors to consider, mainly to the methods and the reporting of the findings. 

  • The methods state: “The level of significance was set as p<0.05. For all tests, Bonferroni-corrected p-values are reported.” From reading the results, my understanding was that the alpha was set as 0.05 for main effects/interactions and then Bonferroni-corrected values are presented for the post-hoc tests. Could the authors please clarify? Further, given the number of comparisons and that the alpha for main/interactions effects is already uncorrected (if I understood correctly above), then perhaps it would be best to avoid describing p = 0.06 and p=0.09 as marginally significant (e.g. section 3.2.6, 3.3.2, 3.3.6).
  • Accuracy and RT of most cognitive tests are being included as outcomes, adding up to many comparisons, and it is not clear whether all are necessary. For example, what does the accuracy measure in the TMT add that isn’t already included in the overall completion time? Since more mistakes = a longer completion time anyway, was there a hypothesis for why both accuracy and RT of all variables should be examined?
  • It would be great to have some more information about the intervention programmes: What was the adherence to the training programme like? Did this vary between groups? Were there any measures of change in physical function or strength? Is there any information of baseline physical function/health?
  • Given the wide age range (20-61), would it be possible to test whether age moderated the observed cognitive transfer effects?

Minor

  • Could the significant cognitive transfer effects also be shown in a figure?
  • To facilitate future meta-analyses, could the authors include exact p-values in the results/table please.
  • Are there some more up-to-date references that can be used in the introduction? Most citations are 10+ years old, and it could be beneficial to show that recent literature still supports the claims being made.
  • It would be helpful to convey in the introduction that the effects of physical training on cognition have not been consistent, perhaps citing one/some of the meta-analyses indicating null results (e.g. Young, Angevaren & Tabet 2015).

Author Response

  • The methods state: “The level of significance was set as p<0.05. For all tests, Bonferroni-corrected p-values are reported.” From reading the results, my understanding was that the alpha was set as 0.05 for main effects/interactions and then Bonferroni-corrected values are presented for the post-hoc tests. Could the authors please clarify? Further, given the number of comparisons and that the alpha for main/interactions effects is already uncorrected (if I understood correctly above), then perhaps it would be best to avoid describing p = 0.06 and p=0.09 as marginally significant (e.g. section 3.2.6, 3.3.2, 3.3.6).

Response: It was specified in the methods section that the Bonferroni correction was used for all post-hoc test and contrast analyses. The description of the results was modified and the term “marginally significant” was removed.

  • Accuracy and RT of most cognitive tests are being included as outcomes, adding up to many comparisons, and it is not clear whether all are necessary. For example, what does the accuracy measure in the TMT add that isn’t already included in the overall completion time? Since more mistakes = a longer completion time anyway, was there a hypothesis for why both accuracy and RT of all variables should be examined?

Response: The Trial Making Test can be processed both quickly and accurately, and both measures can have independent significance. For example, retest effects are seen on the TMT-A in terms of processing time, but not accuracy. The same is true for many other tests. Since we do not hypothesize which measure is more sensitive to time and training, we would prefer to evaluate and also report both parameters.

  • It would be great to have some more information about the intervention programmes: What was the adherence to the training programme like? Did this vary between groups? Were there any measures of change in physical function or strength? Is there any information of baseline physical function/health?

Response: Adherence to the program - regardless of the group - was very high, as this always took place under controlled conditions with specialized personnel. In the area of physical functions and subjective health, it was determined in advance that all participants had neither health impairments nor serious illnesses.

  • Given the wide age range (20-61), would it be possible to test whether age moderated the observed cognitive transfer effects?

Response: Thanks for the hint. Although the age range is almost equally distributed in all three groups, the middle-aged subjects are more strongly represented overall than the younger and older subjects in each group (age group below 29: Physical Training n=8, Combined Training n=8, Active Control n=6; age group 30-44: Physical Training n=17, Combined Training n=19, Active Control n=20; age group over 45: Physical Training n=8, Combined Training n=8, Active Control n=8). Therefore, we do not think that adding age (e.g., as a possible covariate) would be appropriate and allow for reliable conclusions about age effects.

Minor

  • Could the significant cognitive transfer effects also be shown in a figure?

Response: Since the (now exact) values are already included in the table and the text, we consider a diagram redundant and would rather refrain from providing a further figure.

  • To facilitate future meta-analyses, could the authors include exact p-values in the results/table please.

Response: Exact p-values are now included in table 3 and the result section.

  • Are there some more up-to-date references that can be used in the introduction? Most citations are 10+ years old, and it could be beneficial to show that recent literature still supports the claims being made.

Response: We have added more recent work to the literature.

  • It would be helpful to convey in the introduction that the effects of physical training on cognition have not been consistent, perhaps citing one/some of the meta-analyses indicating null results (e.g. Young, Angevaren & Tabet 2015).

Response: Thank you for this hint, these aspect was added.

Round 2

Reviewer 3 Report

The authors have addressed most of my comments. Regarding adherence to the intervention, the authors have replied to my comment noting that adherence was "very high across groups". I think it would be helpful to have numeric information on this included in the text (e.g. % of sessions completed).

Author Response

The authors have addressed most of my comments. Regarding adherence to the intervention, the authors have replied to my comment noting that adherence was "very high across groups". I think it would be helpful to have numeric information on this included in the text (e.g. % of sessions completed).

Response: It was added in the text that subjects who did not complete the training were excluded from the analysis. However, data from the majority of participants could be analyzed. Overall, 83.34% of the combined training group, 80.49% of the physical training group and 82.93% of the active control group successfully completed the program. Of these, all completed the training according to the specifications, which was monitored by experienced trainers.